# Alphonse II of Aragon (1164–1196)

**Marta Serrano-Coll**

Department of History and Art History, Universitat Rovira i Virgili, 43003 Tarragona, Spain; marta.serrano@urv.cat

**Definition:** Alphonse II King of Aragon (1164–1196). He was the first king of the Crown of Aragon and son of the Queen Petronila of Aragon (1157–1164) and the count of Barcelona, Ramon Berenguer IV (1137–1162). Aware of the new political reality that he embodied as King of Aragon and Count of Barcelona, Alphonse II made significant changes to his iconography. Among the most important of these is the binomial that he incorporated to his pendent seals; that is, a portrayal of Alphonse enthroned as king on the obverse and Alphonse as count and mounted on a horse on the reverse. As a known bibliophile and as a result of his desire to reorganise his chancellery following the union of various political entities, he ordered the compilation of the *Liber Feudorum Maior*, the folios of which demonstrate his *potestas regia* through their lavish iconography. He was no less innovative in his coinage, on which he included, for the first time, the image of his head wearing the crown.

**Keywords:** royal images; royal iconography; kings of Aragon; Crown of Aragon; Alphonse II of Aragon

## 1. The Creation of the So-Called Crown of Aragon

Ramon Berenguer IV, Count of Barcelona and Prince of Aragon, died on 6 August 1162. He had been the de facto ruler of the kingdom following the agreement signed with King Ramiro II on 11 August 1137, which led to his marriage to Princess Petronila.

In 1164, Petronila made a de jure donation of the kingdom of Aragon (of which there is a miniature in the *Liber Feudorum Maior*) to her first son in an event of paramount importance for the kingdom of Aragon and for the county of Barcelona insofar as it united both political entities in a confederate system under the rule of the same sovereign, Alphonse II, nicknamed the Chaste or, because of his predilection for the arts, the Troubadour. For the first time, the count was also king, a legal combination that was resolved by giving pre-eminence to the royal title. Well aware of the relevance and significance of his person as the latest link in the royal chain, and in order to make visible the continuity of the *regia stirps* that he represented, he revived the name and, to great effect, the *signum regis* used by Alphonse I, the last and legendary de facto king of Aragon (about Alphonse II, see: [1–6]).

## 2. Appearance and Cultural Interests

After a minority of eleven years, the young king, whom the documents define as thoughtful and meticulous, reached the age of majority. At the age of 16 he was involved in two important ceremonies: his investiture as a knight and his marriage, the latter a matter of the first order for the interests of the dynasty. Although engaged to the Portuguese *Infanta* Mafalda, on 18 January 1174 he married the *Infanta* Sancha, sister and aunt of the kings of Leon and Castile respectively. Soon, the best Occitan and Catalan troubadours became part of life at court, where they were magnificently welcomed and influenced the king's education. The fragments of songs and poems preserved, written by supporters and opponents of the king, present him as a man whose behaviour does not merit the nickname of chaste, with the only reason for such a title perhaps being that he was not known to have had any bastard children. Nevertheless, he did put his amorous skills into practice and recorded them in manuscripts that show a simple and clear style, typical of the *trobar leu* [7].

These compositions also reveal some of the physical features of the king, who is described as tall and slender and, by his opponents, as lazy, cowardly, impolite and disloyal or, by his supporters, as courteous, noble, benign, liberal and faithful.

He promoted various initiatives, many of them of a religious nature. He encouraged the military orders, supported the veneration of various saints (such as Saint Valerius and Saint Raymond), and commissioned buildings, such as the imposing cathedral of Tarragona and the Carthusian monastery of Scala Dei, the first of this order to be established in the Iberian Peninsula. He was buried in the monastery of Santa Maria de Poblet (Tarragona), where one of his sons was a monk and a member of the Cistercian Order, which Alphonse protected so well, turning the monastery into the repository of his body and his memory, and to which he bequeathed, among other possessions, his royal crown.

### 3. Iconography on Coins and Seals

*3.1. Coins: Survival and Innovation*

Alphonse II was the first to hold the title of King of Aragon and Count of Barcelona in his own right and began by quickly issuing low-grade coins that continued to feature the diademed profile bust/processional cross type. The fact that his dominion extended over different territories meant that his coinage diverged and increased at the same time as his territories expanded: he minted coins in Barcelona, Aragon and Provence, where he had defeated the Counts of Toulouse (Figure 1).

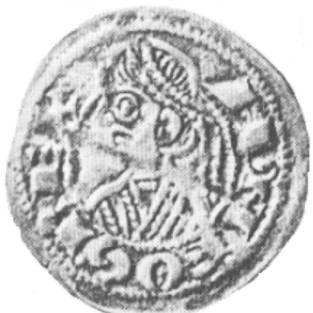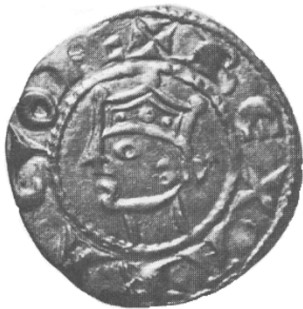

**Figure 1.** *Dineros* of Alphonse II. Aragon and Provence. Obverse. Published by Crusafont, *Acuñaciones de la Corona Catalano-Aragonesa y de los reinos de Aragón y Navarra*, Vico: Madrid, Spain, 1992, p. 211, nums. 141 and 139.

In 1186, he introduced a new type at the Marseilles mint for circulation in Provence known as the royal *diner*, the obverse of which showed a bust of the king in profile wearing, for the first time, a splendid crown. The innovation may be due to the fact that he wanted to mint his own iconography to replace the mitre, the only symbol on the obverse of the coin until then shared by the Count of Provence and the Archbishop of Arles: traditionally, the sovereign of Provence was depicted with the obverse legend REX ARAGONE, which surrounded the episcopal symbol of the mitre. On the reverse, a large cross was surrounded by PRO-VI-NC-IA [8]. This type, which remained unchanged until James I, was the profile bust, which was well known to Alphonse II as it had been used by him and his predecessors in the territories of kingdom, and was now supplemented with a crown, the insignia par excellence of the royal title of Aragon. Perhaps the choice was also due to his relations with his counterparts in Castile and Leon, who minted coins of the same type, perhaps having imported the custom from England [9].

Thus, both in Aragon and Provence, Alphonse is frequently seen with the type of bust in profile on the left, although in Aragon he wore a diadem and headdress with a topknot, as did Peter I and Alphonse I in some of their pieces [10] (and, for more on the hairstyle, see [11]), whereas in Provence he wore a crown richly decorated with precious stones and large pearls. Their reverses maintained the traditional regional types: in Aragon, a

processional cross, and in the territories belonging to the Count of Barcelona, a large cross across the surface.

*3.2. New Pendant Seal with Dual Entitlement*

Alphonse II introduced the pendant seal with the king enthroned on the obverse and in equestrian pose on the reverse (Figure 2), a type that was to endure until Alphonse V (1416–1458), if we do not count the seal of Peter II in which the iconographies are inverted, although he absence of this typology with the names of Ferdinand I or Johan II does not mean that they did not issue them. On the obverse, framed by + SIGILLVM ILDEFONSI REGIS ARA...NENSIS, the king, on a throne and seated on one cushion and with his feet on another cushion, wears a short tunic covered by a cloak tied around his right shoulder, which is uncovered. He wears a crown with three fleurons, a lily on his left and a sword raised in his right hand. On the reverse, the equestrian image is surrounded by the lettering COMITIS BARCHINONENSIS MARCHIONIS ...VINCIE [12] and is similar to that of his father Ramon Berenguer IV, although he is now shown wearing lambrequins.

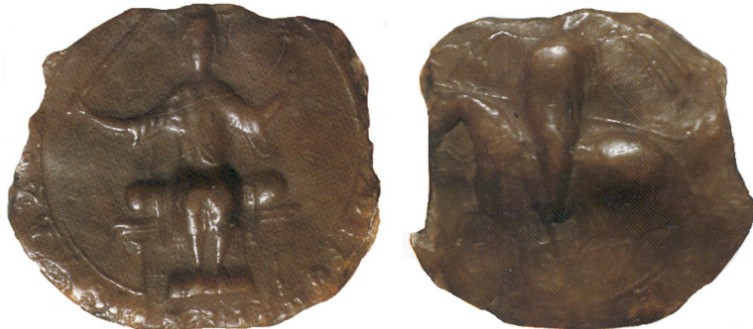

**Figure 2.** Wax seal of Alphonse II (no date). Obverse and reverse. Published by *Reyes de Aragón*; Centellas R. (coord.), Ed., p. 69.

The seal's peculiarity with respect to those of the rest of the Peninsula is that its two faces offer different iconographies. It was a type already used by foreign kings, who had devised it to reflect their possession of more than one title, a reality personified by Alphonse II, king and count at the same time. William the Conqueror had used such a strategy in 1069 in order to demonstrate his own double title: on one side he was shown equestrian as Duke of Normandy and vassal of the King of France, and on the other he was shown on the throne as King of England [13]. Although Alphonse II may have benefitted from another almost contemporary and closer model with the same iconography, namely that of Louis VII the Younger (little used: see Dalas, M. Les sceaux royaux et princiers. Étude iconographique. In [13], pp. 49–337, n. 67, p. 147), the iconographic concomitances between the English and the Aragonese models suggest the influence of the former, which would be reinforced by the links between the Plantagenets and the Aragonese in the territories of southern France [14].

## 4. Iconography in Legal Documents

*4.1. Alphonse II's Confirmation of Privileges*

An early drawing (Figure 3) depicts Alphonse's confirmation of privileges in April 1174, whereby he agreed to ratify those granted by his predecessors to the cathedrals of Huesca and Jaca, and to donate to Bishop Esteban and the diocese of Huesca the tithe of the coinage and the royal revenues (Archivo Diocesano, Huesca, 2–16) [15]. It is the first and only portrait of a king of Aragon in Romanesque miniature, and it is intended to represent the sovereign and thus validate the document it illustrates, which explains why he points to the text with his right hand [16]. Although novel in the Crown of Aragon, it is not the first occurrence in the Iberian Peninsula, as is illustrated, for example, by one of the folios of the *Libro de los Testamentos*, preserved in Cabildo de la Catedral Metropolitana, Oviedo,

also known as *Liber Testamentorum* (Details on the origin of this iconography in [11], pp. 110–111).

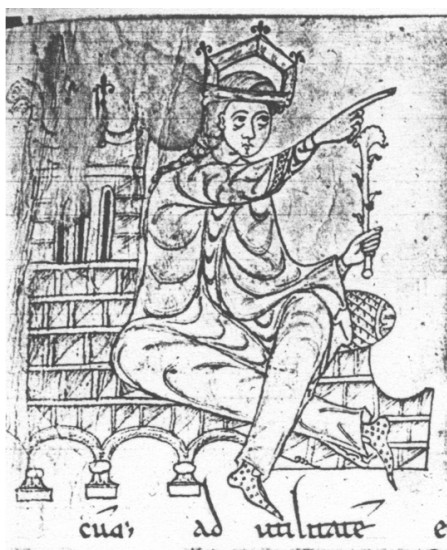

**Figure 3.** *Carta de confirmación de privilegios*, ACH. 2–16, Huesca, Archivo de la Catedral ©. Alphonse II. 1174. Published by Domínguez, J. *Miniatura*. In *Ars Hispaniae*, Plus Ultra: Madrid, Spain, 1958, fig. 6.

The drawing, in black ink, is clearly meant to bolster the sovereign's prestige because it is inserted at the beginning of the document in which it appears and in the position usually occupied by the chrismon, the traditional monogram of Christ that headed important documents in the medieval period [17]. His hair is tied back at the nape of his neck in a plait, which was common practice from the 12th century and into the 13th century, and he wears a visible crown topped with lilies, while in his left hand he holds a flower stem as if it were a sceptre. He is dressed in a smock and cloak and pointed boots, and sits on a cushion on a throne that resembles a fortress, with courses of masonry supported by arches in the manner of a portico and a backrest of a tower with three openings and crowned with fantastic heads and lilies.

*4.2. Liber Feudorum Maior*

The *Liber Feudorum Maior* (Arxiu de la Corona d'Aragó, Barcelona. Canc. Reg. 01) was compiled in Barcelona as a result of the reorganisation of the palatine chancellery following the union of Barcelona and Aragon. It can be dated precisely because the identity of its main copyist, Ramón de Sitges, is known. From 1179 to 1192 and under the orders of Ramón de Caldes, he was responsible for 46 of the 80 folios that have survived [18,19]. A dating for the iconography is more difficult to specify, although most believe it to date to the end of the 12th and beginning of the 13th century.

This cartulary is one among many ambitious undertakings of the court of Alphonse II because it brings together the records relating to the territorial lordship of the kings of Aragon and counts of Barcelona, and demonstrates, in text and image, the *potestas regia*. Its miniatures allude to the textual content, most of which consists of documents that are tributes and agreements between Alphonse II or his predecessors and other kings in the Iberian Peninsula.

The hands of two artists can be seen, the principal one who worked on the book at the time of its creation and another from a different tradition and with a limited chromatic range. It is to the main artist that we owe the two splendid full-page miniatures, another 39 vignettes with scenes of homage, and illuminations of great interest that capture a greater narrative of ceremonial and courtly intention [20–22].

### 4.2.1. The Illustrations of the Early Master

Repetitive and less skilled, this craftsman illuminated several folios plus seven other parchments without text that form part of the group discovered by the director of the Archive of the Crown of Aragon a few years ago and which only offer ornamentation, thus showing that his work predates that of the amanuensis. He drew scenes in which the participants were either agreeing on a pact or marking a situation of vassalage.

Fol. 19r (Figure 4) shows two enthroned sovereigns with their insignia of power and holding hands; they are Alphonse II of Aragon and Alphonse VIII of Castile, who agreed to provide each other with mutual assistance by signing the Treaty of Saragossa in 1170 (see [20], p. 353). The similarities between the two figures, who are both enthroned and dressed in identical clothing, makes it difficult to identify them, although the most significant thing is their common gesture, as they shake hands in the centre of the picture as a sign of the pact they have just signed. Similar representations are not found in other codices of the period, nor in later ones (details in [11], pp. 125–129).

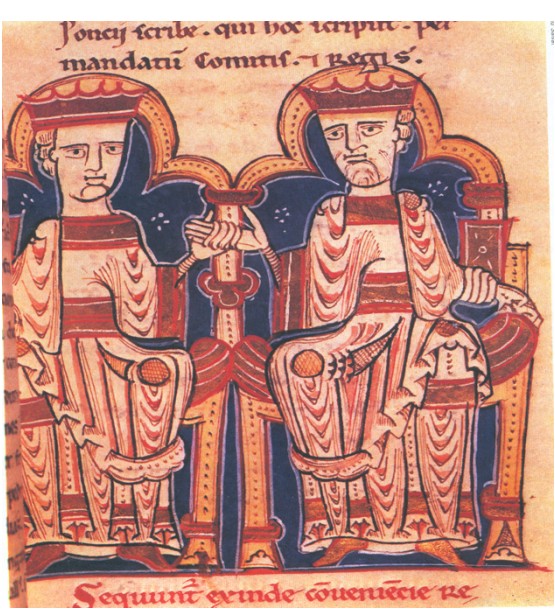

**Figure 4.** *Liber Feudorum Maior*. Canc. Reg. 1, Barcelona, Arxiu de la Corona d'Aragó ©. Fol. 19r. Alphonse II and Alphonse VIII of Castilla. End of the XII Century. Published by Magkanas et alii, *Estudio de las miniaturas y del texto del Liber Feudorum Maior*. Universitat de Barcelona: Barcelona, Spain, 2018, fig. 2.

### 4.2.2. The Illustrations of the Later Master

This second craftsman depicts the palatine atmosphere and reflects scenes of a narrative level and dynamism unparalleled in Romanesque art in the Iberian Peninsula. One of the most spectacular is in fol. 1r (Figure 5), where the dean of Barcelona, Ramon de Caldes, reads some documents to the king in front of six courtiers. They are represented against the background of a series of ailes that vary in colour and descend in height from the central one, and they are beneath a harmonious architectural framework dotted with towers, canopies and battlements: it is, perhaps, a room of the palace adjoining the wall [23]. The bearded Alphonse II, with his hair tied back at the nape of his neck, wears a white smock that peeps out from beneath his elaborate clothing, which is studded with stones at the ends and in the centre of the diamond-shaped lattice. The cloak is draped over the left shoulder, decorated with a succession of triple black stippling, also present on the tassels. He wears a crown, decorated with gemstones, a rod of justice without a finial, and sits on a seat carved at the base with four registers with a succession of small arches and complemented by a footstool and two cushions. His gesture is one of accepting the written

document shown to him by Ramon de Caldes (according to the codification of [24]), which is then deposited on the piece of furniture located between them.

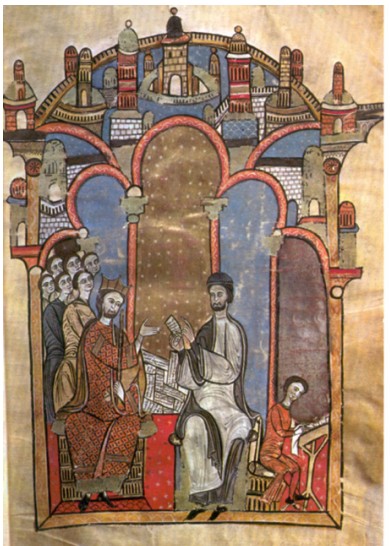 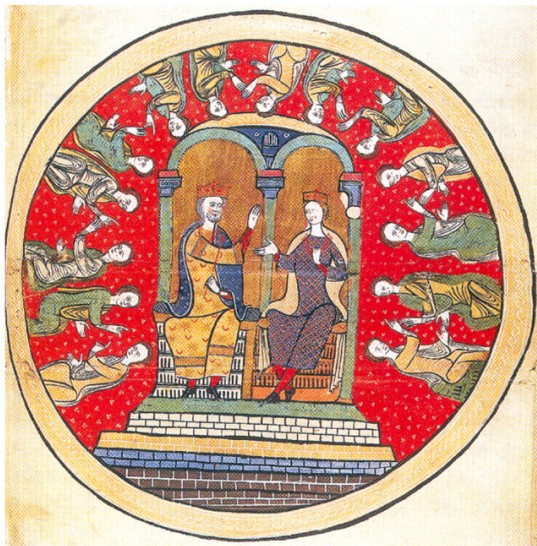

**Figure 5.** *Liber Feudorum Maior*, Canc. Reg. 1, Barcelona, Arxiu de la Corona d'Aragó ©. Fols. 1r and 1st new. Alphonse II and Alphonse II with Sancha? End of the XII Century. Published by Magkanas et alii, *Estudio*, fig. 1 and *Barcelonne-Carvassonne: Destins croisés de deux comtés (IX-XIII siècles)*. Institut Français: Barcelona, Spain, 1997.

Alphonse II is also depicted in other folios, such as 13v, which illustrates his donation of the castle of Cetina to the Knights Hospitallers, or that of 82v, where the king, with his insignia and rich clothing, receives the homage of the nobles of Perpignan in the palace in a beautiful vaulted room crowned by turrets. Some believe that it is an authentic portrait of the sovereign [25], but this must be ruled out as the figure actually shows traces of the "channel style" that is constant in the illustrations by the same artist. We should remember that Channel style had focuses in Corbie, Saint Armand, Marchiennes and Anchin, which reached and became established in various centers in the Iberian Peninsula. An example of this acceptance in Catalonia is the *Viga de la Pasión* (Mueu Nacional d'Art de Catalunya, 15833: see [26]).

It has been proposed that it is Alphonse II who is depicted on the first new folio, exceptional for its circular shape (Figure 5). The circular frame suggests the painting of domes that could have housed royal ceremonials [27], although other possibilities cannot be overlooked, such as the transposition of the iconography of the Hispanic beatus of the Vision of the Lamb, a very important theme of worship throughout the Middle Ages, or that of the profanely decorated gemellions or washbasins, so common in palatial settings in the 13th century, including in the Crown of Aragon.

There are various hypotheses as to the identity of the figures, the most widely accepted being that they are Alphonse II and his wife Sancha [28] (details in [11], p. 132). Both are seated on seats on steps of varying colours. The room, gilded and vaulted, is divided by a double arch that separates the space for the two protagonists. Their hands convey a lively conversation. Alphonse II wears a beautiful crown, wields a slender sceptre and wears, over a red smock that peeps out from under the wide sleeves of his clothes, a beautiful golden cloth with crescents and stars and pearly bands around the edges. On his shoulders is a beautiful blue mantle decorated with white trim and groups of three pearls. The radial arrangement of the figures accompanying them and conversing with each other, paired with the carmine background of the scene decorated with series of three small dots, makes for a beautiful composition. A recent hypothesis, unpublished and very well argued, suggests that it could be Alphonse II and his first-born son, the future Pedro II [29].

### 4.3. Liber Feudorum Ceritaniae

The *Liber Feudorum Ceritaniae* (Arxiu de la Corona d'Aragó, Baracelona. Canc. Reg. 04) may have been completed between 1200 and 1209 and, unlike the *Liber Feudorum Maior*, where the regalia are identical for kings and counts, here, only the kings wear a crown. The book is a compilation of the deeds and documents of the counties of Roussillon and Cerdanya, whose territories formed a single lordship between 1172 and 1276, and, similar to its Barcelona equivalent, it depicts scenes of vassalage, although with notable differences (see [11], p. 134).

In addition to the analogies with the *Life and Miracles of St. Edmund*, from around 1139 (Pierpont Morgan Library, Ms. 736. For more on its iconography, see [30], there are few differences between the depictions of counts and kings, both of whom appear enthroned in a castle-like room with ashlars crowned by towers and battlements. Nevertheless, the artist does emphasise the difference between the two ranks by means of insignia and clothing; for example, on fol. 62r, Alphonse II (Figure 6) is depicted with a visible crown, a precious mantle knotted over his shoulder and rich clothing, while on another folio, a letter that he signs only as count, he is depicted with his head uncovered and in simpler attire. In both cases he stands to the left of the composition and solemnly receives the group of vassals who approach him and hold out their hands for the ceremony of the *immixtio manuum*. In the picture where he appears as king, he wears short clothes decorated with registers of lines and a succession of ornate circles on the inside that contrast with the extreme simplicity of his subjects' attire. He is covered by a red cloak knotted over his right shoulder, leaving his right side free. His crown stands out for its profuse decoration on the rim and finials, hinting at ruby cabochons and incisions, very similar to that on the coin he minted in Provence.

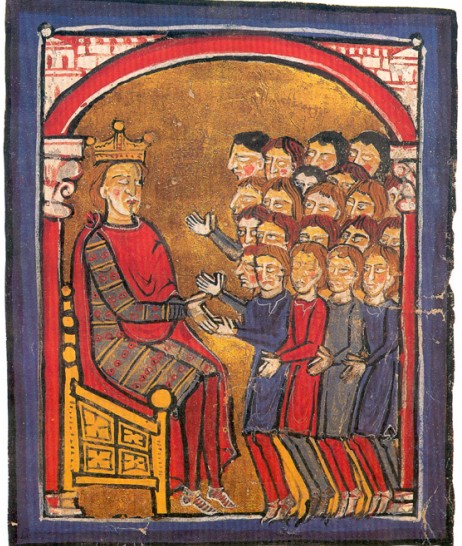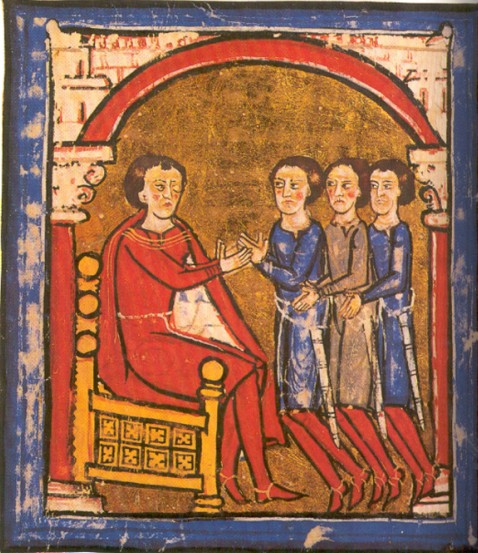

**Figure 6.** *Liber Feudorum Ceritaniae*, Canc. Reg. 4, Barcelona, Arxiu de la Corona d'Aragó ©. Fols. 62r and unknown. Alphonse as king of Aragon and as count of Barcelona. Published by *Catalunya medieval*, pp. 203 and 100.

### 5. Images after His Reign

Other images of Alphonse II can be found throughout the Middle Ages in the Crown of Aragon, such as in the copies of the *Fuero Latino de Teruel*, from the mid-13th century and preserved in Biblioteca Nacional, Madrid, ms. 690, D-44 and Archivo Municipal, Teruel: see [11], pp. 135–137), which show images of the king as an author; some of the folios of the *Usatges i constitucions de Catalunya* from the Arxiu Municipal de la Paeria, in Lleida (ms. 1345) [31,32], dated to the 1320s, on whose fol. 25r the king presides over the Courts of Peace and Truce, or folio 75r, in one of whose initials he appears as author at the

approval of the Courts of Monzón. He also appears in the *Primer Llibre Verd*, from around 1333–1343 (Arxiu Històric de la Ciutat, Barcelona, L.8) (see [23], p. 73, and [33]), whose fol. 88r shows his bust with formal French characteristics; in the *Tercer Llibre Verd*, from around 1342–1348 (Riera puts it back to 1343, see [34]), where the participation of Ferrer Bassa and his workshop or collaborators is evident [35–38], and on folio 49v which features an extraordinary scene of homage; in the *Llibre de privilegis de Cervera* (Arxiu Històric de la Ciutat, Cervera) which was decorated around 1360 and whose first numbered folio presents Alphonse II and James I as the promulgators of the laws they head [39]; and the well-known *Rotlle genealògic de Poblet*, before 1409, one of whose portrait miniatures features Alphonse II, the first offspring to emerge from the double portrait representing his primogenitors, Ramon Berenguer IV and Petronila (study of its illuminations in [40,41]). Of particular note in the field of sculpture is his recumbent (now restored) in the monastery of Poblet, commissioned around 1370 at the behest of Pedro IV. In reality, it was originally sculpted for the tomb of James I [42], which shows the scant value that was given to this type of portrait figure at the time, at least in the Crown of Aragon.

## 6. Conclusions

Alphonse II, aware of the reality he embodied in the same person as both the King of Aragon and the Count of Barcelona, initiated important changes in the way he was represented. Among the most significant, and linked to his dual status, he incorporated for the first time the binomial on his two-faced stamps that would be accepted by all his successors; that is, on the obverse he was the enthroned king, and on the reverse he was an equestrian figure, albeit with the horse concealed. As a result of his desire to reorganise his chancellery, he ordered the illustration of the exceptional *Liber Feudorum Maior*, whose folios demonstrate his *potestas regia*. He was no less innovative in his coinage, where he included the crown on his head for the first time, a measure that highlights the seeming lack of importance given to symbols of sovereignty in the kingdom of Aragon, at least until then. His wife, Sancha, also had an impact in terms of artistic patronage and royal iconography by founding the monastery of Sigena, which was to become a pantheon, and by preserving a possible bifacial wreath inspired by characteristic male models, thus expressing her relationship with power, a practice relatively frequent elsewhere in Europe but which was exceptional in the Iberian Peninsula.

**Funding:** This research was funded by *Edificis i Escenaris religiosos medievals a la Corona d'Aragó*, [2017 SGR 1724]. Generalitat de Catalunya-AGAUR.

**Conflicts of Interest:** The author declares no conflict of interest.

**Entry Link on the Encyclopedia Platform:** https://encyclopedia.pub/16851.

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
