# Peer review of "Alphonse II of Aragon (1164–1196)"

_encyclopedia, doi:10.3390/encyclopedia1040087_

Round 1

Reviewer 1 Report

The text presented by Marta Serrano on Alfonso II of Aragon (1164-1196) can be published with the scientific content that it presents.

The text offers answers to all the questions that we were originally asked: presentation and context of the monarch in his reign; physical appearance, personal and intellectual profile, actions as a promoter; iconographic representation: on seals, coins, manuscripts. All of these sections follow the model of the texts already written by M. Vagnoni.

The whole explanation of the iconography of Alfonso II of Aragon (1164-1196) is completely original. There is no other publication that explains this subject. Marta Serrano has other publications on monarchs from the chronology of 1214, therefore later. The text is therefore original in its scientific content, based on very scattered bibliographical references on the subject. The synthesis and explanation offered by Marta Serrano will be much appreciated. I can affirm that the selection of works is adequate: there is no lack of significant works, nor are there too many.

The explanatory writing is excellent, and offers the reader the necessary information and essential reference bibliography. The figures are also representative, and fits the text.

For all these reasons, I consider the scientific content to be impeccable and excellent.

The text needs to be corrected by a native translator. The text cannot be published without this correction.

 Lidsey, my assessment, if it is to be published, also requires proofreading by a native translator....

Author Response

First of all, thank you very much for the attention with which you have read my work. I will take all your considerations into account. I wanted to say that, of course, English will be checked by a mother tongue when text will be finished adding comments of peer reviewers.

Thank you for your time

Reviewer 2 Report

I particularly enjoyed the original study focusing on the two distinct iconographies of the character, as King of Aragon and Count of Barcelona, on seals, illuminated manuscripts, and coins.

Author Response

Thank you really much, also for your time